# Symptomatic Parapelvic Cysts in Children: Anatomical and Histological Features, Diagnostic Pitfalls and Urological Management

**DOI:** 10.3390/jcm11072035

**Published:** 2022-04-05

**Authors:** Jean-Baptiste Marret, Thomas Blanc, Andre Balaton, Sandro La Vignera, Guido Zanghì, Henri Bernard Lottmann, Vincenzo Bagnara

**Affiliations:** 1Department of Paediatric Surgery and Urology, Hôpital Necker Enfants Malades, APHP, Université de Paris, 149 Rue de Sèvres, 75015 Paris, France; jbaptistemarret@yahoo.fr (J.-B.M.); thomas.blanc@aphp.fr (T.B.); henri.lottmann@aphp.fr (H.B.L.); 2Mechanisms and Therapeutic Strategies of Chronic Kidney Disease, INSERM U1151-CNRS UMR 8253, Institut Necker Enfants Malades, Département “Croissance et Signalisation”, Hôpital Necker Enfants Malades, Université de Paris, 149 Rue de Sèvres, 75015 Paris, France; 3Department of Pathology, Praxea Diagnostics, 1 Rue Galvani, 91300 Massy, France; andre.balaton@praxea-diagnostics.com; 4Department of Clinical and Experimental Medicine, University of Catania, 95123 Catania, Italy; 5Department of General Surgery and Medical-Surgical Specialties, University of Catania, 95123 Catania, Italy; gzanghi@unict.it; 6Department of Paediatric Surgery, Policlinico “G.B. Morgagni”, Via Del Bosco 105, 95125 Catania, Italy; vincenzobagnara@gmail.com

**Keywords:** kidney, parapelvic cyst, UPJ obstruction, paediatric urology, surgery

## Abstract

Background: Symptomatic parapelvic cysts (PPC) are rare entities. Our objective is to highlight specific features of PPC to avoid a misdiagnosis of UPJ obstruction. Methods: We retrospectively reviewed the records of children managed between 2012–2017. Results: All four patients (18 months–8 years) presented with acute renal colic with a large intra-sinusal liquid mass (42–85 mm) on ultrasound, evoking a diagnosis of UPJ obstruction. On preoperative renal scintigraphy (*n* = 3) there was no dilatation of the renal pelvis and ipsilateral differential function was impaired in 2. Diagnosis of PPC was suspected preoperatively in three children (CT scan (*n* = 1); MRI (*n* = 2)) and made peri-operatively (*n* = 1). Preoperative retrograde pyelography (*n* = 3) and a further intraoperative retrograde pyelography with methylene blue (*n* = 1) did not identify communication with the cyst. No renal pelvis was identified in two patients. De-roofing of the cyst was curative in all cases at 5 years mean follow-up (no leakage, cyst recurrence or loss of function) and all 4 patients became asymptomatic after surgery. Histology demonstrated a single flat epithelial cell layer. Renal function normalized in one patient but remained impaired in the other. Conclusion: In case of symptoms of UPJ obstruction with a medial renal liquid mass on ultrasound, PPC should be considered when no dilatated pelvis on renal scan is identified. In such cases, a complementary imaging work-up is mandatory prior to surgery.

## 1. Introduction

The terms “parapelvic or peripelvic cysts” are generally used to describe cysts which lie directly adjacent to the renal pelvis and renal sinus [1,2]. In contrast, a simple renal cyst develops within the renal parenchyma [3,4]. A caliceal diverticulum that arises from a calyx, is usually located peripherally within the renal parenchyma, lined with transitional cell epithelium and communicates with the collecting system [5,6].

In most cases, parapelvic cysts (PPC) are asymptomatic, and are incidental findings on ultrasonography. Regular follow up is recommended [7]. However, in rare cases, the cyst compresses the collecting system and/or renal vessels and becomes symptomatic. These symptoms usually mimic those of UPJ obstruction, which is the main differential diagnosis [8,9].

Our objective was to highlight specific characteristics of symptomatic PPC to avoid a misdiagnosed UPJ obstruction and to describe our management of this rare entity.

## 2. Patients and Methods 

We retrospectively reviewed the medical records of patients managed for a symptomatic PPC between 2012 and 2017 in 2 institutions by 3 surgeons. The clinical characteristics, imaging work-up, surgical management and histological findings were recorded. All patients with a renal cyst or caliceal diverticulum were excluded. The study was conducted in accordance with the French legislation, the Good Clinical Practice and the Declaration of Helsinki.

## 3. Results

All four children with symptomatic PPC presented with recurrent acute episodes of abdominal pain. Patient’s data are summarized in Table 1. Physical examination was normal, including blood pressure, except in patient #3 with a palpable mass in the lumbar region. 

Abdominal ultrasound demonstrated in patient #1 a severe dilatation of what was considered to be the left renal pelvis without significant caliceal dilatation (Figure 1A). In patient #2, it revealed a grossly dilated left renal pelvis (85 mm APD) consistent with a diagnosis of UPJ obstruction. In patient #3 and #4, a large cystic mass was identified.

MRI or CT scan enabled the diagnostic of PPC in patient #2-3-4, demonstrating a very large intra-sinusal cystic mass adjacent to the collecting system (Figure 1B).

A MAG 3 renal scan was performed in 3 children and revealed delayed drainage and impaired function with compression of the renal parenchyma by a cystic lesion in 2 children. One patient also performed a DMSA renal scan which demonstrated the non-enhancing imprint of the cyst (Figure 1C).

Child #1 was operated for symptomatic intermittent ureteropelvic junction obstruction, but a large PPC was discovered per-operatively with a non-dilated renal pelvis and a patent UPJ.

In the 3 other patients, retrograde pyelography confirmed the absence of dilatation of the pelvis and calyces, extrinsic compression of the collecting system and non-opacification of the cyst.

In two cases, renal pelvis was not visualized (Figure 2A,B), the calyces communicating directly with the proximal ureter.

In all cases, de-roofing of the cyst was performed (Figure 2C).

The preoperative retrograde pyelography in three cases and, in one case, a further intraoperative retrograde pyelography with methylene blue, did not identify communication with the cyst (Figure 2D). Histology of the cyst wall demonstrated a single layer of cubic or flat cells.

Recovery was uneventful and the children remained subsequently asymptomatic (mean follow-up of 5 years (3.5–7)). In patient #3, post-operative US revealed an 85 mm caliceal dilatation of a lower calix, secondary to stenosis of a caliceal root with no recurrence of the PPC.

## 4. Discussion

Surgical management is indicated in the very rare symptomatic PPC. Most published cases have been reported in adults. Marsupialization of the cyst via open surgery was the classical approach [10]. Alternative treatment options have been reported more recently such as ureteroscopic marsupialization of the cyst in the renal pelvis [11,12] and de-roofing of the cyst by either a transperitoneal or retroperitoneal laparoscopic approach [7,13].

The purpose of our report, the largest series of symptomatic parapelvic cysts in children, was to define the characteristic features of PPCs and to highlight the diagnostic pitfalls which may lead to a delayed or misdiagnosis of UPJ obstruction.

So far, there have been very few cases of PPC reported in children. In 1980, Chan reported an 8-year-old boy with symptomatic PPC (haematuria and hypertension). Following de-roofing of the cyst, the haematuria resolved, and the blood pressure normalized [2]. Patel published another case report of symptomatic PPC in a six-year-old girl with intermittent renal colic. The cyst was partially excised. No communication between the cyst and urinary tract was identified and histology revealed a single layer of predominantly cuboidal cells [9]. Lakhoo et al. reported a case of a 10-year-old boy with recurrent abdominal pain and pelvicaliceal dilatation on US, suggestive of UPJ obstruction. However, no pelvic dilatation was observed on intravenous urography. A renal scan demonstrated normal drainage and differential function. Surgery was undertaken with a preoperative diagnosis of intermittent UPJ obstruction. Perioperatively, a large (60 × 75 mm) thick-walled cyst adherent to the pelvis in the hilar region was identified. Subtotal excision was performed. There was no communication between the pelvicaliceal system and the cyst. Although this mass was described as a caliceal diverticulum, it fulfilled all the criteria of a PPC [14]. In 2006, Dobremez reported the case of a 2-year-old girl with hypertension due to PPC. Elongation of the calyces was observed on intravenous urography. Enucleation was performed. Postoperatively, the blood pressure normalized [1].

Symptomatic PPC mimics intermittent acute UPJ obstruction. Ultrasound alone may lead to misdiagnosis, particularly when there is a discordance between a major dilatation and a good drainage on renal scan. It stresses the importance of a complete imaging work-up with MRI or CT-scan and retrograde pyelography.

The risk of severe and irreversible impairment of renal function justifies surgical management. It consists of an initial retrograde pyelography followed by subtotal de-roofing of the cyst. A laparoscopic approach is a safe alternative to open lumbotomy or subcostal flank incision. An attempt to excise the portion of cyst wall adherent to the renal sinus and renal vasculature in the region of the hilum has previously led to complete loss of the kidney [15].

Interestingly, a communication between the cyst and the collecting system has neither been shown in the literature nor in our cases. In two of our cases, the anatomy of the upper urinary tract was highly unusual with an absence of renal pelvis, a finding that has not been previously reported.

There has been no urine leakage, cyst recurrence or significant loss of renal function in any of our four patients, thus validating de-roofing as the appropriate surgical management.

Of note, one patient had a post-operative large caliceal dilatation of a lower calix, secondary to stenosis of a thin caliceal root without recurrence of the PPC.

Some authors have attributed the origins of PPC to lymphatic ectasia [8], whereas other authors have suggested that they are embryological remnants arising from the mesonephric duct [9]. Our histology finding of single flat epithelial cell layer suggests that PPC and the urinary tract are structures from independent origin.

Finally, with regard to painful symptoms, we hypothesize that the cause could be linked to the extrinsic compression that the cyst can cause on the renal excretory path, creating a momentary stasis especially in the case of hyperhydration and thus resulting in renal colic. This hypothesis is supported by the scintigraphic data that demonstrates, in two cases, accumulation of the radiotracer in the renal pelvis, which in any case empties spontaneously and, even more, after administration of a diuretic.

Although limited to four cases, due to the extreme rarity of the condition, this report contributes to a better understanding of the anatomical and histological characteristics, diagnosis, and appropriate management of symptomatic PPC.

Our report underlines the importance of considering the diagnosis of PPC, a very rare condition, in the case of atypical or contradictory imaging in a patient with clinical features otherwise suggestive of an UPJ obstruction. In such cases, a complete imaging work-up is of upmost importance in the evaluation of the upper urinary tract anatomy.

## Figures and Tables

**Figure 1 jcm-11-02035-f001:**
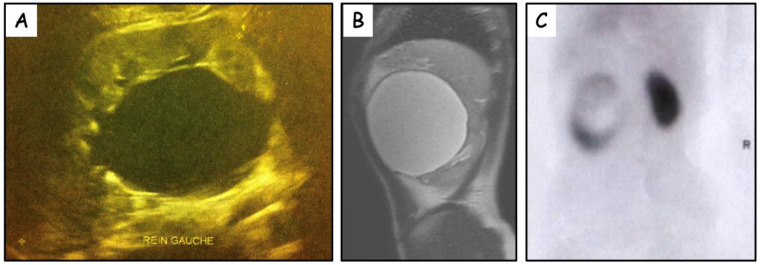
(**A**) Renal ultrasound. Large fluid-filled structure mimicking a grossly dilated renal pelvis (42 mm antero-posterior diameter). No caliceal dilatation. (**B**) Pre-operative MRI (sagittal view) demonstrating very large cystic lesion adjacent to the left renal pelvis. (**C**) Pre-operative DMSA renal scan demonstrating photopenic lesion (cyst) occupying the central zone of the left kidney.

**Figure 2 jcm-11-02035-f002:**
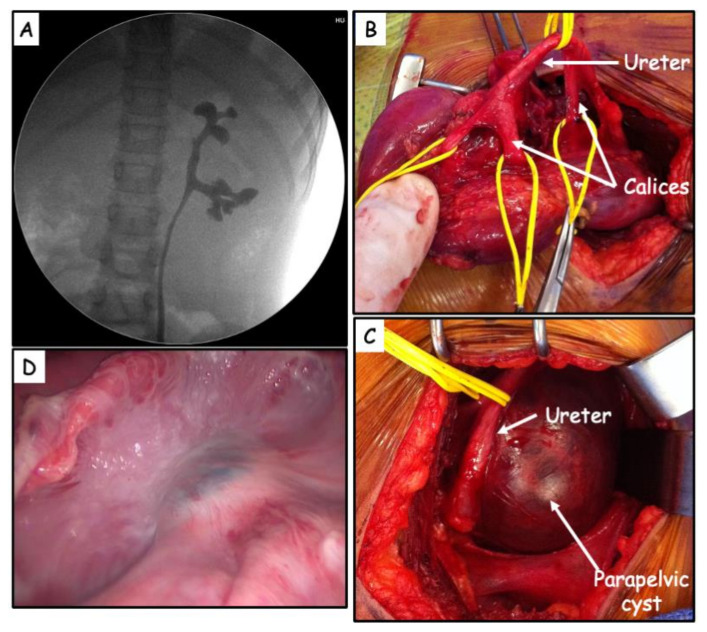
(**A**) Absence of renal pelvis demonstrated on retrograde pyelography. (**B**) Renal anatomy after subtotal excision of the cyst. No identifiable renal pelvis. Calyces communicating directly with the proximal ureter. (**C**) Large parapelvic cyst causing compression. (**D**) Intraoperative retrograde pyelography with methylene blue (visible in transparency in a renal calyx in the center of the photo). No communication was identified between the unroofed cyst and the renal pelvis.

**Table 1 jcm-11-02035-t001:** Demographics, patients’ characteristics, management and outcomes of symptomatic parapelvic cysts.

Case #	Age	Presentation	Renal US	Size (mm)	MRI/CT Scan	Pre-op MAG3 RS	Approach	Preoperative Retrograde Pyelography	Intraoperative Retrograde Pyelography Methylene blue	Procedure	Follow-up (years)	Outcome	Post op MAG3 RS
1-M	18 months	Abdominal pain Vomiting	Pelvis dilatation	42	-	48%	Subcostal flank incision	-	-	De-roofing	7	Asymptomatic	-
2-F	8 years	Abdominal pain	Pelvis dilatation	85	PPC	38%	Laparoscopy	No pelvicalyceal dilatation	No communication	De-roofing	4	Asymptomatic	51%
3-M	7 years	Renal colic Vomiting	Cystic mass	55	PPC	21%	Subcostal flank incision	No renal pelvis	-	De-roofing	5.5	Asymptomatic	17%
4-F	5 years	UTI Abdominal pain	Cystic mass	60	PPC	-	Laparoscopy	No renal pelvis	-	De-roofing	3.5	Asymptomatic	-

UTI: urinary tract infection; PPC: parapelvic cyst; US: ultrasound; RS: renal scan.

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
