# Peer review of "Symptomatic Parapelvic Cysts in Children: Anatomical and Histological Features, Diagnostic Pitfalls and Urological Management"

_jcm, 2022, doi:10.3390/jcm11072035_

Round 1

Reviewer 1 Report

none

Author Response

attached file

Reviewer 2 Report

The authors reported the specific features of symptomatic parapelvic cysts (PPC). Though PPC is an extremely rare condition, they successfully found that the case of symptoms of UPJ obstruction with a medial renal liquid mass on ultrasound, PPC should be considered when no dilatated pelvis on renal scan is identified.

Well written.

This is an interesting and impressed report. The contents are concise and easy to read.

This paper has the potential to be applied in the future.

Author Response

Answers to questions

Questions

Correction/Comments

1

Do the authors mean nuclear renography?

On preoperative renal scintigraphy.

We have changed the text.

2

Did all patients undergo retrograde pyelography? Did all patients receive methylene blu? Was it given intravenously?

Preoperative retrograde pyelography (n=3) and a further intraoperative retrograde pyelography with methylene blue (n=1) did not identify communication with the cyst.

We have changed the text and corrected table 1 which was incorrect

3

Did the symptoms resolve?

all 4 patients became asymptomatic after surgery.

We have changed the text.

4

The form of renography is a bit unclear. Table 1 shows that 3 patient had MAG3 renography but the only renography image is labelled DMSA. Was DMSA renography done in more than one patient?

You are right, we wanted to insert an image of a DMSA scan, done by only one of the 4 patients, to show what it looks like in case of a parapelvic cyst.

We have changed the text.

5

Did the cyst cause any impairment in drainage of the ipsilateral kidneys?

Yes. It is already reported in the text. Now having added the radiotracer it should be clearer. See points 4, 8

6

Table 1 states that two patients underwent lumbotomy. Was this a true dorsal lumbotomy in the prone position? Access to the renal hilum is limited in this approach. Were either done via a subcostal flank incision?

You are right. The incision was a subcostal flank in both cases.

We have changed the text.

7

The absence of a renal pelvis is very rare. It would nice to see the retrograde pyelogram in the other patient with no renal pelvis in order to educate the reader.

Unfortunately, editorial needs prevent us from adding more photos. However, if the manuscript will be accepted, you can contact us and we will be happy to send it to you

8

Figure 1 shows a DMSA scan but DMSA scans are not mentioned in the text. The text says “ A renal scan was performed”. The radiotracers used would be instructive to the reader.

You are right.

We have changed the text. See point 4

9

Figure 2 has interesting images. Labels like on Fig 2C would be helpful for Fig 2D

We can not quickly change the photo. However we have changed the legend to better explain the photo

10

Was methylene blue injected intravenously or directly into the collecting system?

We have changed the text

See also points 2, 9 and table 1

11

It would be helpful if the authors hypothesized the cause of the pain. Were the cysts compressing the collecting system and caused upper urinary tract obstruction. Or was the pain due to pressure within the renal parenchyma caused by the enlarging cysts

Finally, with regard to painful symptoms, we hypothesize that the cause could be linked to the extrinsic compression  that the cyst can cause on the renal excretory path, creating a momentary stasis especially in the case of hyperhydration and thus resulting in renal colic. This hypothesis is supported by the scintigraphic data that demonstrates, in two cases, accumulation of the radiotracer in the renal pelvis which in any case empties spontaneously and, even more, after administration of a diuretic.

We have added this hypothesis to the discussion

Reviewer 3 Report

This case report of four cases of symptomatic parapelvic renal cysts in children from two institutions is the largest such report.  The authors present the cases fairly well.

Abstract-

Do the authors mean nuclear renography?

Did all patients undergo retrograde pyelography?  Did all patients receive methylene blue?  Was it given intravenously?

Did the symptoms resolve?

Introduction

Relatively short but there is not much literature to discuss

Methods

The form of renography is a bit unclear.  Table 1 shows that 3 patient had MAG3 renography, but the only renography image is labelled DMSA. Was DMSA renography done in more than one patient?  Did the cyst cause any impairment in drainage of the ipsilateral kidneys?

Table 1 states that two patient underwent lumbotomy.  Was this a true dorsal lumbotomy in the prone position? Access to the renal hilum is limited in this approach.  Were either done via a subcostal flank incision?

The absence of a renal pelvis is very rare.  It would nice to see the retrograde pyelogram in the other patient with no renal pelvis in order to educate the reader.

Figure 1 shows a DMSA scan, but DMSA scans are not mentioned in the text.  The text says "A renal scan was performed..."  The radiotracers used would be instructive to the reader.

Figure 2 has interesting images.  Labels like on Fig 2C would be helpful for Fig 2D.

Was methylene blue injected intravenously or directly into the collecting system?

Discussion

It would be helpful if the authors hypothesized the cause of the pain.  Were the cysts compressing the collecting system and caused upper urinary tract obstruction.  Or was the pain due to pressure within the renal parechyma caused by the enlarging cysts.

Author Response

(The authors gave the same response as above.)
